# Selective Retrieval of Individual Cells from Microfluidic Arrays Combining Dielectrophoretic Force and Directed Hydrodynamic Flow

**DOI:** 10.3390/mi11030322

**Published:** 2020-03-20

**Authors:** Pierre-Emmanuel Thiriet, Joern Pezoldt, Gabriele Gambardella, Kevin Keim, Bart Deplancke, Carlotta Guiducci

**Affiliations:** 1Laboratory of Life Sciences Electronics, École Polytechnique Fédérale de Lausanne, 1015 Lausanne, CH, Switzerland; gabriele.gambardella@epfl.ch (G.G.); kevin.keim@epfl.ch (K.K.); carlotta.guiducci@epfl.ch (C.G.); 2Laboratory of Systems Biology and Genetics, École Polytechnique Fédérale de Lausanne, 1015 Lausanne, CH, Switzerland; jorn.pezoldt@epfl.ch (J.P.); bart.deplancke@epfl.ch (B.D.)

**Keywords:** single-cell microfluidics, single-cell recovery, single-cell array, hydrodynamic trapping, electrokinetics, tridimensional electrodes, dielectrophoresis (DEP), mRNA sequencing, Drop-seq

## Abstract

Hydrodynamic-based microfluidic platforms enable single-cell arraying and analysis over time. Despite the advantages of established microfluidic systems, long-term analysis and proliferation of cells selected in such devices require off-chip recovery of cells as well as an investigation of on-chip analysis on cell phenotype, requirements still largely unmet. Here, we introduce a device for single-cell isolation, selective retrieval and off-chip recovery. To this end, singularly addressable three-dimensional electrodes are embedded within a microfluidic channel, allowing the selective release of single cells from their trapping site through application of a negative dielectrophoretic (DEP) force. Selective capture and release are carried out in standard culture medium and cells can be subsequently mitigated towards a recovery well using micro-engineered hybrid SU-8/PDMS pneumatic valves. Importantly, transcriptional analysis of recovered cells revealed only marginal alteration of their molecular profile upon DEP application, underscored by minor transcriptional changes induced upon injection into the microfluidic device. Therefore, the established microfluidic system combining targeted DEP manipulation with downstream hydrodynamic coordination of single cells provides a powerful means to handle and manipulate individual cells within one device.

## 1. Introduction

The analysis of single cells is routinely carried out by means of in-flow measurements such as optical marker detection in flow cytometry. Despite the high throughput that can be achieved with those techniques, they only provide a snapshot of cells’ properties at a certain time point with limited possibility to observe their behavior over time [1,2]. Cell arraying, i.e., the separation and localization of individual cells or doublets on a surface, allows observation of cells over extended periods of time, collection of secreted biomolecules [3] and recording of the response to specific stimuli [4]. Observation systems based on cell arraying are unique tools used to unveil mechanisms of cell-to-cell interaction [5], polarized cell proliferation [6], in vitro fertilization [7], etc.

Parallel immobilization of cells can be obtained through various methods—for instance, cell sedimentation on microwells [8], localization of single cells by means of optical tweezers [9], trapping by dielectrophoretic cages [10,11], segregation into small chambers sealed by PDMS valves [12] or hydrodynamic trapping [13].

Hydrodynamic trapping is a method based on the immobilization of cells of a specific size range at various locations of a microfluidic channel. Trapping sites are defined by tight side-wall openings of low fluidic resistance where single cells are led to by the laminar flow. Cells captured with this method are continuously exposed to a flow of medium, allowing for delivery of nutrients and disposal of waste. Previous works characterizing hydrodynamic cell-trapping systems have reported a trapping efficiency—defined as the percentage of traps filled after injection of cells—between 75% and 99%. These methods have been used to capture multiple cell types [14] and, in some cases, to localize rare cells [15]. They also proved to be valuable tools for investigation of single-cell behaviors. For instance, Dura et al. [5] could conjointly place single dendritic cells and T-cells and measure heterogeneity in the activation of T-cells.

Furthermore, the association of the molecular state of a cell with its on-chip characterization could help unveil new cellular mechanisms. Kimmerling et al. [16] compared intra- and inter-lineage transcriptomes within a cell population by capturing multiple generations of a single starting cell in subsequent traps. In that study, the analysis of the transcriptome was performed upon the retrieval of the entire lineage from the chip. The release of cells from their hydrodynamic traps was carried through application of a backflow pushing the cell out of the trap. This approach was also used by Kim et al. [12], who trapped unicellular microalgae in PDMS chambers for on-chip culturing and selectively retrieved the content of one chamber with a sophisticated three-layer PDMS valving system. Yeo et al. [15] combined centrifugal and hydrodynamic forces to isolate circulating tumor cells from a mixed cell population; to enable release, each trap was connected to an independent backflow channel. This method is suitable for collection of extremely rare cells such as circulating tumor cells, but it suffers from very low throughput and poor scalability, as the number of cells that can be retrieved is limited by the number of backflow channels that can be placed on the chip. Tan et al. [17] could retrieve cells encapsulated in hydrogel beads, creating an air expansion on the trap site generated with laser heating. This platform is limited by the complexity of the setup and the damage to the cells that may be induced by heat.

These proof-of-concept technologies underscore the necessity and also the challenges in combining continuous observation of cells on chip with further off-chip investigations. Dielectrophoresis (DEP) forces localized on the trap site by implementing microelectrodes in its vicinity could be utilized to control cell trapping in very high-throughput arrays. In fact, electric signals can be easily multiplexed, while keeping a very small footprint of the DEP actuator. Zhu et al. [18] used such forces to selectively release single yeast cells from their trapping site. The yeasts, placed in synthetic low-conductivity medium [19], were not recovered from the chip. A possible cause for this could be the well-known challenge of fabricating valves combined with microfluidics fabricated with stiff materials [20].

It is important to mention that a potential issue of DEP electrokinetic actuators could be cellular stress resulting from the application of polarizing electric fields. The impact of DEP signals on a population of cells in a single large chamber has been investigated by Nerguizian et al. [21] and by Flanagan et al. [22], although no study has dissected transcriptional changes induced by exposing cells to DEP forces applied identically to the whole population and on single cells. Additionally, to date, no studies have addressed the impact of DEP application and general effects of off- and on-chip handling on the molecular state of the cell [15,16].

In this study, we introduce a Microfluidic Platform for Arraying and Release of single Cell (MiPARC). This DEP-based platform allows for selective trapping and off-chip recovery of individual mammalian cells in their native culture medium. Selective release of T-lymphocytes was achieved by three-dimensional DEP actuators, integrated for the first time in SU-8 channels with a width of 25 µm. In order to obtain a precise handle on the flow in the microfluidic branches of the chip, and thus to recover single cells, we developed a PDMS valve technology compatible with SU-8 microfluidics, overcoming one of the main limitation of devices using walls made of stiff materials. We investigated the stress induced on cells by our platform through analysis of the molecular phenotype via mRNA sequencing, revealing no impact of DEP application on the transcriptional signature of the cells, super-seeded by minor alterations of the cellular molecular state introduced by hydrodynamic forces within the microfluidic system.

## 2. Materials and Methods

### 2.1. Device Fabrication

Our device consists of sixteen microfluidic traps arrayed inside a tree-like structure, as shown in Figure 1. The two valves located upstream allow control of liquid injection in the chip while the two valves located downstream enable the recovery of single cells. The height of the microfluidic channel as well as of the electrodes is 15 µm.

#### 2.1.1. Fabrication of the Microfluidic Chip

Our system is based on a glass substrate. A detailed picture of the process flow is reported in Appendix A. After sputtering of a layer of Ti/Pt/Ti (20/200/20 nm) on the wafer (Pfeiffer Spider 600, Pfeiffer Vacuum, Asslar, Germany), planar metal lines are patterned through standard photolithography and ion beam etching (Veeco Nexus IBE 350, Veeco, Plainview, NY, USA). In order to isolate electrical lines from the liquid, a 300 nm layer of oxide is sputtered on the wafer. This layer is then etched (SPTS APS dielectric etcher, SPTS Technologies, Newport, UK) in the region where the pillars will be connected to the planar metal lines. Cylindrical vertical pillars of SU-8 (Microchem 3025, Microresist Technologies, Berlin, Germany) are patterned on the exposed planar electrodes. Then, a thin layer of Ti/Pt (20/200 nm) is sputtered to cover the entire wafer with metal. This metal layer is then removed everywhere except on the pillar walls through vertical ion beam etching. In a successive process step, microfluidic channels of the same height of the electrodes are patterned in SU-8, hence allowing a micrometer-scale alignment of the electrodes inside the microfluidic system (Figure 1c). The most critical step of this process flow is the development of this second SU-8 layer that creates the microfluidic structure. In fact, an appropriate development time is crucial to avoid SU-8 residues clogging the fluidic restriction (insufficient development) and excessive development of the SU-8 that would result in apertures larger than designed and a delamination. With the chosen parameters (development time of 4′30′’ in PGMEA: Propylene glycol methyl ether acetate), we recorded a 95% yield.

#### 2.1.2. Fabrication of the PDMS Top Layer and Valves

The PDMS valving system placed on the top of the microfluidic channel is composed of two layers: a control layer made of a thick PDMS layer and a thin membrane that will be deflected in order to close or open the microfluidic channel (for more details, please refer to Appendix A. The fabrication of the molds for the control layer is carried out with standard photolithography and dry etching (AMS 200 Alcatel) on a silicon wafer. PDMS prepared at a ratio of 5:1 is poured over the silicon mold which has previously been treated with trimethylchlorosilane (TMCS). The membrane is fabricated by spin-coating of PDMS at a ratio of 5:1 on a TMCS-treated silicon wafer, the speed of rotation is 2500 rpm and results in a thickness of 15 µm. Both this PDMS membrane layer and the lid PDMS layer are partially cured at 80 °C for 30 min and then aligned and bonded at 80 °C for 1.5 h. Finally, the PDMS coverslip is treated with 3-aminopropyl triethoxysilane (APTES) and irreversibly bonded by incubating the system at 150 °C for 2 h. This step presented some experimental challenges, since the alignment of both the fluidic and control channels is carried out manually. Additionally, pressing the chip with the PDMS coverslip with too much pressure might lead to the adhesion of the membrane with the bottom of the channel before the baking step, resulting in irreversible bonding between the membrane and the microfluidic channel after curing, preventing the valve form operating properly. Nonetheless, a yield above 80% could be obtained for the whole fabrication and packaging process.

### 2.2. Finite Element Simulations

Finite element simulations of the device were carried out using COMSOL Multiphysics 5.3 (Comsol Inc., Burlington, USA). The main objectives of these simulations were to optimize the dimensions of the microfluidic channels in order to achieve efficient hydrodynamic trapping of cells and to describe the electrical field in the trapping region to evaluate the best geometry to maximize the DEP force obtained. The simulation results reported in Figure 2 were all conducted on a 3D model. The flow velocity in the channels was simulated using the Laminar Flow module and a normal mesh, the pressure difference between inlet and outlet was set at 0.1 mbar and a no slip condition was applied on the fluidic walls (Figure 2a,b). The electric field was simulated using the Electrostatics module and a fine mesh, the voltage difference between both electrodes was set at 10V, which corresponds to the amplitude that will be later used for cell actuation (Figure 2c). All computations were computationally inexpensive and could be conducted in less than 10 min.

### 2.3. Cell Preparation

The human T-lymphocytes Jurkat cell line was cultured in suspension in a Roswell Park Memorial Institute (RPMI) medium supplemented with 10% fetal bovine serum and 1% antibiotics (L-Glutamine-penicillin-streptomycin). Prior to application of cells to the microfluidics chip, cells were washed two times in phosphate-buffered saline (PBS) and resuspended in RPMI medium at a concentration of 300,000 cells/mL.

### 2.4. Cell Injection and Recovery

Prior to injection, the chip was primed with RPMI medium to coat the internal channel surfaces with albumin. All injections were carried out using a pressure system (OB-1, Elveflow (FR)). Hybrid SU-8/PDMS valves were filled with ultrapure water and closed applying a pressure of 3500 mbar. Cells were injected at a pressure of 0.5 mbar into the channel. Trapping of a cell can be monitored with an inverted microscope (Leica DMIL) equipped with a CMOS camera (Leica DFC295). Once a cell is trapped hydrodynamically, it can be released by applying an electrical signal via a signal generator (Agilent 33220A) with a frequency ranging from 1 to 20 MHz and an amplitude ranging from 8 to 10 Vpp. The signal was addressed specifically to each single electrode through a multiplexing system implemented on a custom made PCB and controlled by a Python interface (Appendix A). After release, the valves were configured to allow guided release of cells to the recovery chamber that was priorly filled with 5 µl of RPMI medium.

### 2.5. Sequencing Methodology

#### 2.5.1. Library Preparation

Recovered cells (approximately 400) were centrifuged at 600 g for 5 min and the supernatant was replaced with 25 µl of PBS containing 0.01% bovine serum albumin (Sigma, Kanagawa, Japan). Subsequently, 25 µL of lysis buffer was added, containing 0.1% Sarkosyl (Sigma), 5 mM EDTA (Life Technologies, Carlsbad, CA, USA), 0.1 M Tris (pH 7.5, Sigma) and 25 mM 1,4-Dithioreitol (DTT, Sigma), 800 units/mL RNase inhibitor (NEB, Ipswich, MA, USA) and 500 Drop-seq beads (Beads, Lot 120817, ChemGenes, Wilmington, MA, USA). All bead pelleting steps were carried out at 1000 g for 1 min in 1.5 mL microtubes (Axygen, Union City, CA, USA). Reverse transcription (RT), exonuclease I (ExoI) treatment and PCR were performed as described by Macosko et al. with minor adaptations [23]. Lysed cells were incubated at 1400 rpm for 5 min at room temperature and subsequently washed twice with 1 mL of 6x SSC buffer (Sigma). Reverse transcription was performed for 90 min at 42 °C in 50 µl of 1 mM dNTPs (Clontech, Mountain View, CA, USA), 2.5 µM template switch oligo (see Table 1), 1250 units/mL RNase inhibitor, 1x Maxima RT buffer and 10,000 units/mL Maxima H minus reverse transcriptase (Thermo Fisher Scientific, Waltham, MA, USA). Drop-seq beads were washed twice with 0.5% SDS (Applichem, Omaha, NE, USA) in 10 mM Tris (TE-SDS), twice with 0.01% Tween-20 (Sigma) in 10 mM Tris (TE-TW) and once with 10 mM Tris pH 7.5. The Drop-seq bead pellet was then incubated with 50 µl of exonuclease mix containing 1x Exonuclease I Buffer and 1000 units/mL Exonuclease I (NEB) and incubated at 37 °C for 45 min at 1400 rpm). Drop-seq beads were washed twice with TE-SDS, twice with TE-TW and once with double-distilled H_2_O. Beads were amplified by PCR in 25 µL of 1x Hifi HotStart Readymix (Kapa Biosystems, Wilmington, MA, USA) and 0.8 µM TSO-PCR primer (Table 1) at 95 °C for 3 min; 4 cycles of: 98 °C for 20 sec, 65 °C for 45 sec, 72 °C for 3 min; then, 16 cycles of: 98 °C for 20 sec, 67 °C for 20 sec, 72 °C for 3 min and an extension step of 5 min. Libraries were purified using Ampure XP beads (at a ratio of 0.6x to remove small fragments), cDNA was quantified using a Qubit HS kit (Thermo Fisher Scientific) and integrity was analyzed on a Fragment Analyzer (Agilent). Libraries were prepared using in house-produced Tn5 loaded with adapters, as described [24]. Size selected and purified libraries were sequenced paired-end on a NextSeq 500 system (Illumina, San Diego, CA, USA) in High Output mode following recommendations from the original protocol (read 1—20 bp and read 2—50 bp) [23].

#### 2.5.2. Data Analysis and Availability

The data analysis was performed using the Drop-seq tools package on the EPFL SCITAS HPC platform. After trimming and sequence tagging, reads were aligned to the human reference genome (hg38) using STAR (version 2.7.0.e) [25]. Following the alignment, the gene annotation was added, bead synthesis errors were corrected, and cell barcodes extracted. Subsequently, the BAM files containing the processed data were used to obtain digital gene expression matrices. Only cell barcodes with at least 50 UMI (Unique Molecular Identifier) were retained. Downstream data analysis was carried out using R (version 3.5.0), with *DESeq2* (version 1.22.2) [26] for identifying differentially expressed genes in pair-wise comparison. Plots were generated using the R package *ggplot2* (version 3.0.0). The Gene Expression Omnibus (GEO) accession number for the RNA-seq data reported in this paper is GSE143190.

## 3. Results and Discussion

### 3.1. Cell Trapping and Release

Sequential injection of individual cells at the trap location is an important feature of our platform, and it can be ensured by setting the width of the channel in the vicinity of the traps at 25 µm. Single-cell hydrodynamic traps are placed along this channel. Hydrodynamic trapping is a technique based on the use of mechanical restrictions to segregate particles from a main channel. The separation can be carried out efficiently if the flow going through the restriction channel is slightly higher than the flow in the main channel, however, the flow in the restriction should not be too high to avoid trapping of multiple cells. The traps are arranged on the branches of a tree-like fluidic structure shown in Figure 2a. Parallel-channel design is used to restrain possible clogging due to contamination to single branches. Figure 2b shows the finite element simulation of a single trap (COMSOL Multiphysics 5.3) that is composed of two elements: a fluidic bypass along the channel and a fluid path through the trap. The fluidic resistance of the bypass is 1.2-fold larger than that through the empty trap, which leads the cell towards the constriction (Figure 3a,c). Assuming an average diameter of lymphocyte of 10 µm, the height of the channel was set at 15 µm to avoid multiple stacking of cells in the trapping sites. The width of the main channel is 25 µm and the diameter of a cell is approximately 10 µm, resulting in a distance between the electrode extruding from the SU-8 wall and the cell in the trap of 15 µm. This distance has been chosen to allow a cell flowing in the channel after a trapping event to be guided in the bypass channel without risking clogging the whole channel. The number of traps which are filled with single cells upon injection is typically 90%, in agreement with the trapping efficiency values reported in literature [14]. We also measured the probability of a cell to be trapped by an empty trap as 75% in case of T-lymphocytes.

Having achieved targeted single-cell localization in the traps, we next aimed to use electrodes embedded in close proximity to the microfluidic channel to selectively release one specific single cell by means of DEP. Dielectrophoresis phenomena results in the displacement of polarizable particles in a non-uniform electric field. The particle experiences the formation of a dipole—the orientation of which depends on the relative permittivity of both the particle and its surrounding medium. If the particle is more polarizable than the medium, the induced dipole is oriented along the electric field. Reciprocally, the induced dipole is oriented against the electric field if the medium is more polarizable than the particle. Additionally, if the electrical field applied is non-uniform, the particle will move due to a higher field strength on one side of the particle. The electric field generated by the electrodes is constrained in the trap aperture as shown in the COMSOL simulation in Figure 2c. The field gradient is highest in the region of cell trapping. As the cell polarizability is lower than the polarizability of the surrounding medium, the cell is pushed toward the regions of the weaker field, i.e., out of the trap. This repulsive force—called negative dielectrophoretic (DEP) force—permits the targeted release of the cell (Figure 3b,d, Appendix A). The optimization of the operation parameters was carried out both experimentally and theoretically. The frequency was maintained above 1 MHz to avoid electrolysis that could be observed below that threshold and could lead to gaseous species formation. The force applied to the cell due to DEP is proportional to the real part of the Clausius–Mossotti factor according to the following equation:(1)$$F_{DEP}=2\pi R^{3}\varepsilon_{medium}Re\left(CM\right)\nabla E^{2}$$
where *R* is the radius of the cell, $\varepsilon_{medium}$ the dielectric permittivity of the medium, *E* the electric field in the channel and *CM* the Clausius–Mossotti factor.

Importantly, in the case of T-lymphocytes, the real part of the Clausius–Mossotti factor will drastically decrease at frequencies higher than 20 MHz [27]. Consequently, we set a working frequency range between 1 and 20 MHz. The minimal voltage value experimentally-identified to trigger efficient release in our configuration is 8 Vpp. Hence, in order to minimize the impact of application of electrical fields on cells, we maintained the voltage applied to cells between 8 and 10 Vpp throughout this study.

Each electrode is singularly addressable through a printed circuit board (PCB) directing the electrical signals toward the chosen electrodes, allowing a selective release of the cell of interest, as depicted in Figure 3c,d. Furthermore, the negative DEP force can be used in order to prevent trapping of unwanted cells (Appendix A).

The most immediate approach to generate an electric field in a microfluidic system is to place two electrodes at the inlet and exit reservoirs, respectively [28]. The large distance between the electrodes and the active regions where the electrical gradient is required necessitates high voltages [28], which has many drawbacks—for instance, (i) the need to generate high-amplitude AC signals, (ii) the generation of heat and (iii) the induction of water electrolysis phenomena at the electrodes and consequent bubble generation, affecting cell viability.

The integration of electrodes near the trapping regions reduces those issues, since the amplitude of the signal can be drastically reduced to achieve the desired dielectrophoretic force. Planar microelectrodes and on-chip connections have been used to drive electrical signal to the site of interest [18]. However, this approach results in non-uniform electric fields over the channel height, decreasing towards the top of the channel. The employment of side-wall microelectrodes, instead, would generate the repulsive force more efficiently and homogeneously from the bottom to the top of the aperture.

One of the novel features of our approach resides in the replacement of standard planar electrodes by vertical electrodes according to a process previously developed by our group [29], consisting of a conformal coating of SU-8 pillars with metal, to obtain singularly addressable side-wall electrodes with a width of 40 µm and height of 15 µm (Figure 1c). In each cell trap, two vertical electrodes are placed across the hydrodynamic aperture, with the closest to the cell being only 15 µm away from it. This configuration can be successfully obtained only thanks to the high precision of the technology we developed to fabricate vertical electrodes in microfluidic channels. Vertical electrode fabrication has previously been reported through pyrolysis of photoresist structures [30] or metal ion implantation in PDMS [31] but none of those approaches enable such precision in the position of electrodes to ensure their integration inside a microfluidic channel as narrow as the one presented here.

The use of vertical electrodes improves the efficiency of force generation. In fact, we could set the amplitude of the DEP signals to lower values with respect to previous works that employ planar electrodes (10 Vpp of this work versus 20 Vpp [18]). Furthermore, we could afford to maintain the cells in their native medium (RPMI, conductivity 1300 mS/m) for all our experiments, a substantial advantage considering the requirements of cell biologists. Since DEP forces are weakened by high ionic strengths, most designs employing DEP to apply electrokinetic forces substitute the native medium with a synthetic one with lower conductivity [18,21]. Hence, by being able to retain cells in their native culture medium, cellular stress is reduced, washing and centrifugation steps limited and general compatibility with conventional cell-based assays achieved.

### 3.2. Single-Cell Handling for Accurate Retrieval

Our microfluidic system allows the recovery from the chip of a given cell that has been released from its trap. We designed and implemented a novel valve-based design to drive the released cell to a recovery line ending in an open well on the chip.

Microfluidic channels are made of SU-8, an epoxy-based negative photoresist that will not deform upon application of pressure. In order to create valves in the chip, we had to overpass the classical PDMS microfluidic valves, widely known as Quake’s valves [32], which are based on elastomeric PDMS microfluidic structures. Previous studies have proposed solutions to integrate valves within microfluidics that were fabricated with hard material, as in our case. Huang et al. [33] designed valves by means of a PDMS layer sandwiched between two poly(methyl methacrylate (PMMA) channels. Similarly, Lee et al. [20] integrated a PDMS membrane between two polycarbonate (PC) layers.

In this study, we introduce for the first time PDMS-based valves combined with SU-8 microfluidics, a solution that opens new opportunities for lab-on-chip devices offered by the combination of hard photoresist microfluidics and on-chip PDMS valves. Four valves are placed on the microfluidic branches of the structure—the two inlet valves allow switching of solutions to be injected in the main channel, while the two outlet valves enable to switch between the recovery and the disposal channel, as previously mentioned (See Section 2.1.2 and Appendix A for more details). Figure 4a,b show the operation of one of our valves when the application of a high pressure on the control channel closes the valve sufficiently to stop the passage of cells. However, as the SU-8 fluidic cannot deform, the valve is only partially sealed, and a subcellular-sized aperture remains between the SU-8 walls and the PDMS membrane (See Appendix A). The residual flow through the valve is used to drive the cells to the recovery well. Figure 4c,d show the recovery of a single cell leaving the chip and entering the recovery well from which it can be simply pipetted out.

### 3.3. Transcriptional Profiling of MiPARC Processed Cells Reveals Negligible Impact of DEP Application on the Cellular Molecular State

The established MiPARC platform traps single cells in microfluidic constrictions, allowing for their release in the main channel and their selective extraction off chip. As those cells could be further analyzed or potentially expanded for adoptive transfer, the impact of the microfluidics operation and the applied electric field on cell function needed to be assessed. To globally inspect the molecular changes to the cell, we thus analyzed their transcriptome using RNA-sequencing in the framework of DEP manipulation.

In order to determine whether the microfluidics setup or the applied DEP impact the molecular properties, Jurkat T-cells were injected and either passed through the microfluidics setup for an average duration of three minutes in absence or in the presence of DEP forces. After retrieval, cells were collected and cultured off chip for three hours to permit potential alterations by the DEP field or the fluidic forces to be represented transcriptionally. Input cells that were solely cultured served as negative control (Input), whilst cells cultured for three hours under Phorbol-12-myristate 13-acetate and Ionomycin activation (PMA/Iono), globally activating transcription based on protein kinase C (PKC) activation and calcium ion influx [34], served as positive control for global T-cell. Due to the low number of maximally 400 cells per sample and the high volume of up to 100 µl of cell culture medium, we employed mRNA capture beads to obtain the mRNA of the lysed cells and detected over 7600 genes across the experimental conditions (Figure 5a). Importantly, cDNA quality was similar across the conditions with marginally higher yield for “PMA/Iono”-activated cells (Appendix A). Additionally, hierarchical clustering of the detected mitochondrial genes revealed that cells processed with MiPARC showed similar expression intensities as compared to the “Input”, indicating that cells were viable to a similar extent as unprocessed cells (Appendix A). Differential expression analysis identified 114 genes across all pair-wise comparisons. Principal component (PC) analysis on all differentially expressed genes (DEGs) revealed that the first PC1 segregated “PMA/Iono”-activated cells from all other experimental conditions, whereas PC2 separated untreated cells from those subjected to the microfluidics chip. Importantly, there was no defined separation between cells subjected to DEP (Chip-DEP) or solely injected in the chip (Chip-Ctrl) (Figure 5b). The high concordance between “Chip-DEP” and “Chip-Ctrl” was underscored by the observation that no DEG could be detected, whereas 74 or 59 DEGs were identified when comparing “PMA/Iono” cells to “Chip-Ctrl” or “Chip-DEP”, respectively (Figure 5c up). Importantly, the number of DEGs when comparing “Chip-Ctrl” or “Chip-DEP” to the “Input” conditions was substantially lower with 14 or 24, respectively (Figure 5c down). Importantly, genes associated with stress responses such as heat-shock proteins, *HSPA6* and *HSP90AA1*, chaperons like *CLU* and genes involved in stress recovery responses including *DNAJB1* and Ubiquitin (*UBC*) were significantly upregulated only under “PMA/Iono” conditions (Figure 5d left). Interestingly, a minor proportion of genes was upregulated for “PMA/Iono” and both Chip conditions, encompassing predominantly genes involved in cell proliferation including *EGR1*, *SMC2*, *FOS* and *FOSB* or activation *TRAC* and *H3F3B* (Figure 5d left). When comparing the genes consistently differentially expressed between PMA/Iono-activated cells and cells injected into the chip, the vast majority of activation and stress response genes was upregulated solely under “PMA/Iono” condition, whereas CXCR4 expression was only upregulated in cells that were flown through the chip (Figure 5d right).

Based on these transcriptional profiling results, we conclude that the impact of injection into and extraction from the microfluidics chip outweighs the changes wrought by the electric field onto the transcriptional landscape of the cell under the implemented culture conditions. This could be in part due to the very confined DEP, permitting the utilization of a low voltage of 8 Vpp at frequencies of 20 MHz, thereby limiting the extent of transcriptional changes observed in a previous study [21]. Furthermore, it was previously shown that the application of negative DEP, as utilized in the MiPARC system, does not alter the viability or differentiation capacity of neuronal embryonic stem cells, even at long DEP exposure times of up to 30 min [22], which exceeds the pulsed approach used within MiPARC by far. Although the transcriptional alterations instigated within the MiPARC system are minor, as compared to global activation of the cell, and unavoidable when implementing microfluidic cell handling, special care should be taken to minimize stressors such as high pressure or long retention times within the microfluidic devices. Regardless, application of short-term DEP for accurate retrieval of cells does only minorly impinge on the transcriptome at the obtained resolution.

## 4. Conclusions

Here, we presented a fully integrated system for single-cell isolation and off-chip recovery. The device is based on trapping constriction with dimensions that were designed to obtain efficient single-cell arraying. Moreover, hybrid SU-8/PDMS valve fabrication is described and implemented here for the first time to allow control of different flow sources and outputs. We showed for the first time the integration of vertical SU-8 electrodes in a narrow microfluidic channel. SU-8 electrodes were used in this work to enable selective release of single cells from specific traps by means of a negative DEP force generated by low voltage application at the electrodes and achieved in the cell’s native culture medium. After release, cells were successfully recovered off chip and the phenotypic effect of injection in the microfluidic channel and exposure to DEP force were investigated through mRNA sequencing. For the first time, phenotypical effects induced by microfluidics and DEP were characterized separately, showing that the changes triggered by handling the cell in the microfluidic device outweigh the changes caused by the electrical field.

Currently, our device is composed of sixteen traps that can be singularly addressed. The MiPARC can be scaled up to permit the trapping, selective release and recovery of hundreds of cells, thus increasing the throughput of our platform. Indeed, hydrodynamic trap-based systems for single-cell arraying have already been achieved with parallelism of 1000 trapping locations on one chip [12]. Our approach for specific release by means of electrical actuators is compatible with this and even higher scales.

## Figures and Tables

**Figure 1 micromachines-11-00322-f001:**
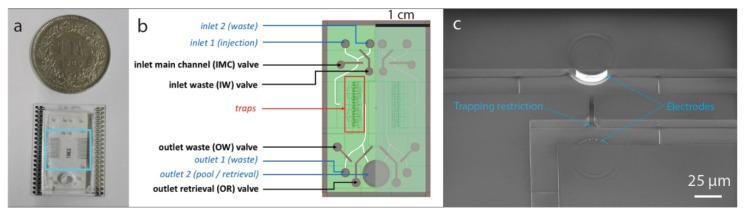
Device layout description and fabrication. (**a**) Picture of the microfluidic chip used for the experiments. The chip hosts two symmetrical parallel channels that share the outlet-well used for cell recovery. The blue box indicates the area where the cells are trapped in hydrodynamic sites. (**b**) Microfluidic layout of the chip. The SU-8 based microfluidic main channels are shown in green, while PDMS control channels are depicted in gray. When a cell enters the chip through inlet 1, it can be directed either towards the traps for immobilization and observation or to the inlet 2 if it needs to be discarded. After release for a trap, a cell can be recovered from the pool at outlet 2 or disposed through the outlet waste. (**c**) Magnified scanning electron microscopy image of a trap with electrodes embedded in the microfluidic channel. The aperture is 5 µm in width and the electrode extrusion from the microfluidic channel measures approximately 7 µm.

**Figure 2 micromachines-11-00322-f002:**
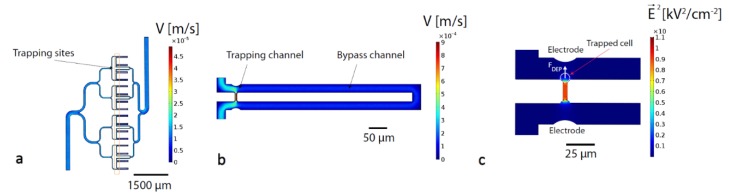
Simulations of the device layout and working principle. (**a**) Finite element simulation of the fluid velocity in the microfluidic channel (Inlet pressure: 0.1 mbar). (**b**) Finite element simulation of the fluid velocity in the microfluidic channel used to determine the ideal length of the channel ensuring the trapping of a single cell (Inlet pressure: 0.1 mbar). (**c**) Finite element simulation of the electric field in the channel. The field gradient is higher in the trapping region of the cell, generating a negative dielectrophoretic (DEP) force sufficient for the release of the cell from the trap (voltage difference between electrodes: 10 V).

**Figure 3 micromachines-11-00322-f003:**
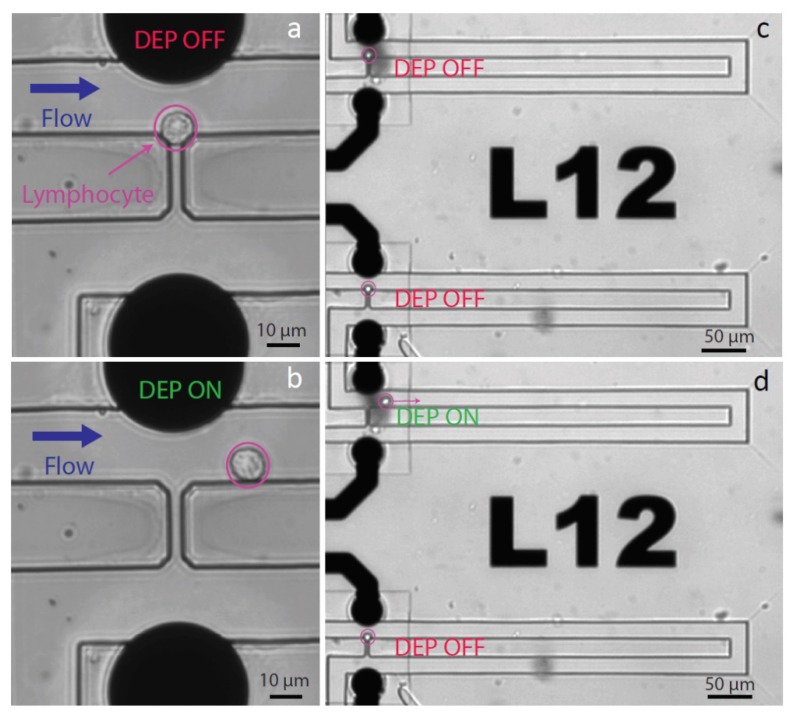
Selective single-cell retrieval. (**a,b**) A single lymphocyte can be trapped in the hydrodynamic constriction (**a**) and gently released (b) through application of negative DEP force activated by 10 Vpp voltage at 10 MHz. (**c**,**d**) The cell at the top is released, while the cell at the bottom is kept in the trap. The release is carried out with a 10 Vpp voltage at 10 MHz. A custom-made printed circuit board (PCB) enables the selective release of a single T lymphocyte. For more details, please refer to the Appendix A.

**Figure 4 micromachines-11-00322-f004:**
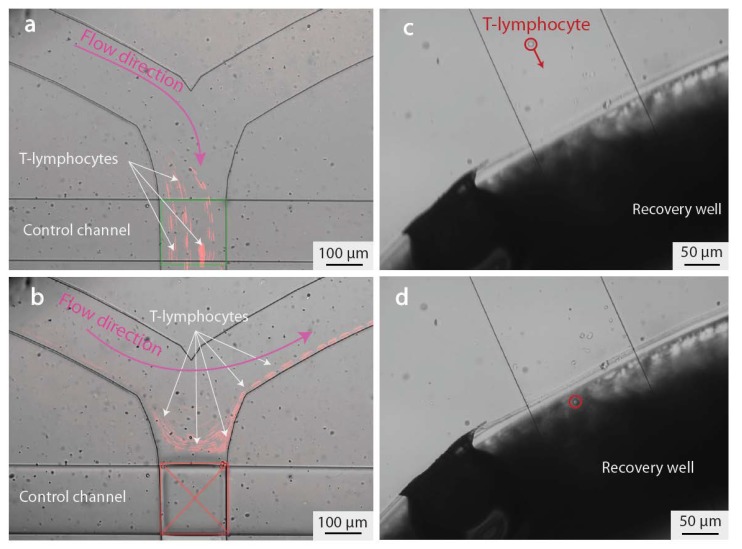
Single-cell recovery. (**a,b**) Operation of a hybrid SU-8/PDMS valve used to control the flow in the microfluidic channel. Trajectories of the lymphocytes in the channel is depicted in red. As SU-8 is stiff, high pressure (3500 mbar) must be exerted on the above PDMS layer in order to close the valve. When the valve is open, all cells are directed towards the main microfluidic channel (**a**), while closing the valve will prevent them from entering the channel (**b**). Combining four of those valves enables control of the injection and recovery of cells from the chip. (**c,d**) A single lymphocyte exiting the chip and entering the recovery channel, where it can be pipetted and further analyzed. This direct recovery in a well prevents the cells from sticking and being lost in outlet pipes. A video showing the trajectory of a cell from the trapping area to the recovery well is available as Appendix A.

**Figure 5 micromachines-11-00322-f005:**
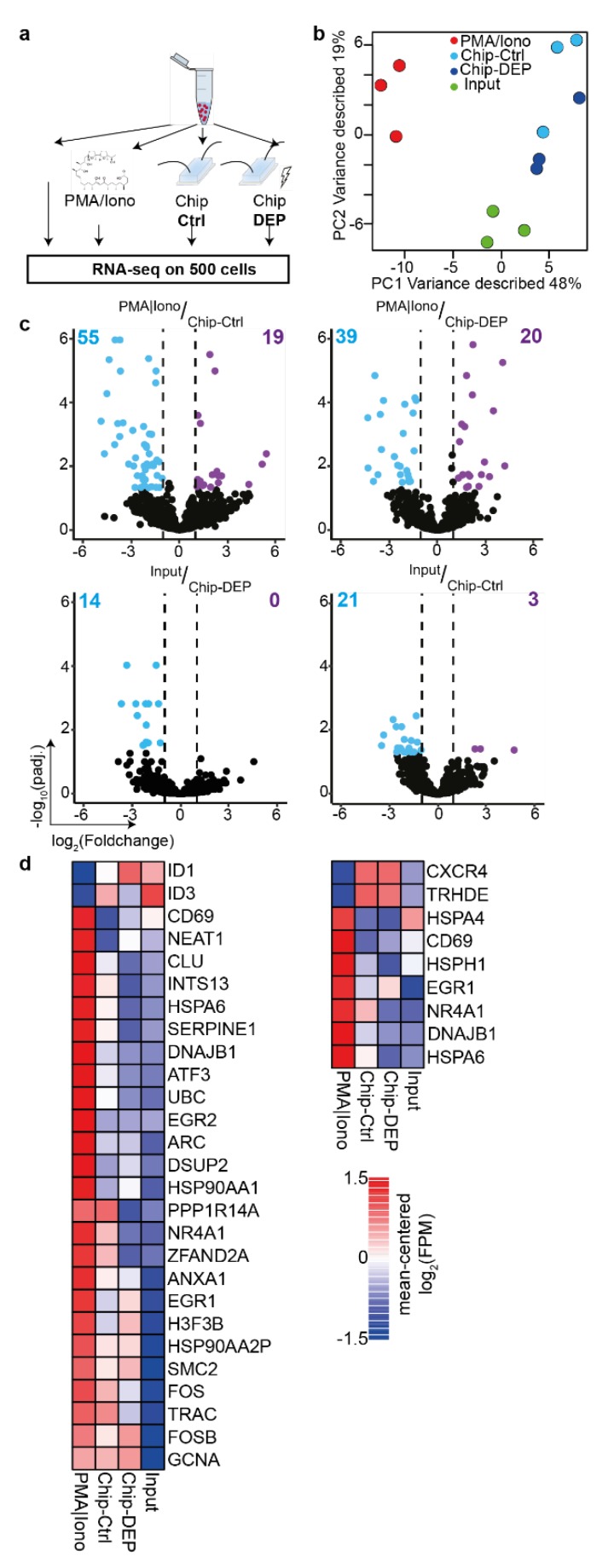
The electric field has minor molecular impact on Jurkat T-cells. (**a**) Jurkat cells were either injected into the microfluidics chip (Chip-Ctrl) or additionally subjected to the electric field used for accurate capture and retrieval of cells (Chip-DEP). Controls were either the input cells (Input) or cells activated for three hours under Phorbol-12-myristate 13-acetate and Ionomycin activation (PMA/Iono). Cells from all conditions were cultured for three hours to permit transcriptional changes to take place subsequent to treatment. (**b**) Principal component analysis on all differentially expressed genes (number of DEGs: 117). (**c**) Volcano plots of mean RNA-seq FPM (Fragment Per Million) comparing indicated samples. Number of DEGs is indicated. (**d**) Heatmaps represent expression of selected DEGs. Left: DEGs between PMA/Iono-stimulated and Input cells. Right: DEGs common on comparing Chip-Ctrl and Chip-DEP to PMA/Iono-stimulated cells. Experiments were performed in three independent biological replicates. DEG, differentially expressed gene (absolute(log2[foldchange] >= 1 and padj. <= 0.05).

**Table 1 micromachines-11-00322-t001:** Primers used for reverse transcription and library preparation.

ID	Sequence
TSO	AAGCAGTGGTATCAACGCAGAGTGAATrGrGrG
TSO-PCR	AAGCAGTGGTATCAACGCAGAGT

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
