# Peer review of "Selective Retrieval of Individual Cells from Microfluidic Arrays Combining Dielectrophoretic Force and Directed Hydrodynamic Flow"

_micromachines, 2020, doi:10.3390/mi11030322_

Round 1
Reviewer 1 Report
This manuscript reports a microfluidic device patterned with three-dimensional (3D) microelectrode arrays for cell trapping and manipulation using dielectrophoresis (DEP). The details about fabrication processes of 3D vertical microelectrodes were presented. The performance of the microfluidic device was evaluated by on-chip capture, release and recovery of Jurkat T cells. Moreover, it has been demonstrated that DEP and hydrodynamic forces do not significantly affect gene expression. This work is important in developing compact, functional devices for on-chip single cell analysis. I would recommend this manuscript to be accepted for publication in Micromachines if the authors are willing to address my comments below:
- I think it would be helpful if the authors can provide some details about the optimisation of operation parameters and device dimensions. For example, why the AC electrical field with magnitude of 10 voltage and frequency of 10 MHz is applied for cell release? Why the gap between two electrodes is set at 15 um? How about the effect of flow rate on cell trapping?
- Can the authors explain the principles and mechanisms of DEP and hydrodynamics for cell release and trapping?
- It is great to see that the authors have evaluated the effects of DEP and hydrodynamic forces on gene expression (Figure 5). I would recommend the authors to compare the viability of cells before and after flow through the system (with and without applying DEP forces). It is important to obtain viable target cells of interest for further analysis.
- I think Figure 3 only shows that DEP force can be used to release target cells in a single location, not selective cell release. In order to demonstrate this, can the authors provide an extra figure to show that DEP forces can be applied at different electrodes and result in the release (DEP on) or trapping of cells (DEP off) at different trap apertures?
- Figure 4a and 4b, can the authors provide relatively clean, high-resolution images (perhaps fluorescent images) to show cell movement with different valve status (on/off)?
- Figure 2, the text/label is too small to read, and it is unclear what do the contours show in Figure 2a.
- I can’t see a reference in main text to Supplemental Video S4.
Author Response
Dear reviewer,
We kindly ask you to look at the attached document to see our replies to your comments.
Best regards,
Pierre-Emmanuel Thiriet

Reviewer 2 Report
Selective retrieval of individual cells from microfluidic arrays combining dielectrophoretic force and directed hydrodynamic flow
Pierre-Emmanuel Thiriet, Joern Pezoldt, Gabriele Gambardella, Kevin Keim, Bart Deplancke and Carlotta Guiducci
Comments:
The authors have presented microfluidic device with 16 hydrodynamic traps to capture cells in a continuous flow, where single cells can be released by a negative dielectrophoretic (DEP) force. The negative DEP can be activated on-demand by an array of 3D electrodes integrated with an SU-8/PDMS hybrid microfluidic device. Additionally, the authors claim to have implemented a first of its kind PDMS-valve with SU-8 channels. The transcriptional analysis shows that the cells exposed to the DEP force were not much different from those flown through the microfluidic channels without application of DEP force. However, both of these cases have induced some changes to the cells compared to a control sample that was not run through the device. The manuscript is very thorough in its explanation of experimental details and could be suitable for publication in the Micromachines journal after the incorporation of following comments:
- It is highly recommended to investigate and describe the effect of applied voltage and frequency on the cell’s transcriptional analysis. It is not clear that how is the effect of DEP force quantified without varying the applied voltage/frequency to investigate the trapping efficiency?
- What is the residence time of a single cell inside the microfluidic channels? How would an extended time effect the cells transcriptional analysis?
- Page 2 of 13, line 80-84: how is the localized DEP effect on cells is different from a single large-scale chamber?
- What exactly is the overall throughput of the device? Considering only 400-500 cells are collected per sample, is it a sufficient sample concentration for most biological analysis?
- How would the trapping sites be realistically scaled up for a considerable number >1000 of cells manipulation?
- Figure 2 and 3: Orientation of different panels is confusing. Please keep the direction of the panels consistent throughout a figure.
- Figure 4: It is very difficult for a reader to distinguish the cells from the background microchannel. It is suggested that the authors may use ImageJ to remove background and use Z-project tool to obtain the cells trajectories.
- Movie 1: what is the average flowrate or velocity inside the microchannel when a cell is hydrodynamically trapped and then released by the negative DEP? Apart from mentioning the pressure value, a flowrate value could be helpful to the reader.
- Movie 2: as the DEP is ON, the continuous flow cell moves past the trap but another cell at the bottom is not moving, why is it so? Please keep the orientation of the movie frames and figures consistent or clarify that some of the channels are mirror image of the others. It is confusing for the reader.
- It is not clear to the reviewer that what has been the role of valves in the whole study and how critical it was?
The authors have designed, fabricated and realized a very sophisticated platform that can be further utilized to perform interesting studies in the future. The current study could still be polished for better understanding of the reader.
Author Response

(The authors gave the same response as above.)

Reviewer 3 Report
Thiriet et al. present a device that combines fluidic trapping with a dielectrophoretic release mechanism to manipulate cells. The project overcame several engineering challenges and is validated through Jurkat transcriptomic analysis. Apart from minor revisions, I recommend publication of the manuscript.
Fig 1. I recommend a more holistic figure 1 that better conveys the motivation of the project. An extended version of Fig S1 that includes a close-up trap schematic/ micrograph could be more suited in that regard. While likely consuming most time to get this right, the current schematic from Fig 1a may be moved to the supplement, detailing process specifics akin to Fig S3. Such a ordering my better reflect the general readers interest to follow the major concepts compared to someone keen a re-building the details that will closely study the supplement..
Methods/Fig S1/2: Please provide CAD files and a bill of materials for both chip and PCB design to enable reproducibility of results. I also recommend to make the python script available to readers. Given the multiple fabrication steps for both chip halves, please comment on respective yields and the major precautions/handling tweaks that helped with increasing those yields.
Methods/Fig2: Please provide sufficient detail on the finite element modeling to allow for reproduction of results, ideally as a separate methods section. Was a 2D or a 3D model used - at what mesh resolution? What boundary conditions were defined for fluid and electrics. What DEP field, frequency/ amplitude was applied (The same as in Fig 3 caption)? How computationally expensive were those models (CPU times)?
line 157: Please specify how many cells were recovered approximately? Are these cases of all 16 traps filled with a single cell and recovered after identical DEP stimulation? A later mention of 400 cells (line 307) may be interpreted in that about 400 cells were present in all the channel volume and only the 16 trapped ones actually exposed to the DEP. Please clarify, if all cells quantified where transiently trapped and then released?
line 253: May low melting solder electrodes or Salt electrodes (PMID 24671446), as used in FADS be used alternatively.
line 307: How long was the DEP stimulation compared to the 3 hr on chip incubation prior to recovery and transcriptomic analysis?
line 365: Quake valves on top of rigid fluidics are not exactly un-precedented: https://iopscience.iop.org/article/10.1088/1367-2630/11/7/075027
Author Response

(The authors gave the same response as above.)

Round 2
Reviewer 1 Report
The authors have addressed all my comments. I would like to recommend this manuscript for publication.
Reviewer 2 Report
The authors have revised the manuscript thoroughly and adequately responded to the reviewer's comments. The manuscript is suitable for publication. The authors may incorporate the following comments in their final revision.
The authors may include this information in the revised manuscript:
- "The average time spent by the cell in the microfluidic channels is around three minutes."
- "The actual throughput of the platform is about 250 cells per hour, which is not sufficient for most biological analysis implying screening of thousands of cells."
-
"Considering a speed of the flow in the channel of 10 μm/sec we can derive a flow rate in the 25 μm channel of 180 pL/min which also correspond to a flow rate of 1.4 nL/min in the main channel. The flowrate value has been added to all the trapping related videos."